# Preparation and Characterization of Photocatalytically Active Antibacterial Surfaces Covered with Acrylic Matrix Embedded Nano-ZnO and Nano-ZnO/Ag

**DOI:** 10.3390/nano11123384

**Published:** 2021-12-14

**Authors:** Merilin Rosenberg, Meeri Visnapuu, Kristjan Saal, Dmytro Danilian, Rainer Pärna, Angela Ivask, Vambola Kisand

**Affiliations:** 1Institute of Molecular and Cell Biology, University of Tartu, Riia 23, 51010 Tartu, Estonia; merilin.rosenberg@ut.ee; 2Laboratory of Environmental Toxicology, National Institute of Chemical Physics and Biophysics, Akadeemia tee 23, 12618 Tallinn, Estonia; 3Department of Chemistry and Biotechnology, Tallinn University of Technology, Akadeemia tee 15, 12618 Tallinn, Estonia; 4Institute of Physics, University of Tartu, W. Ostwaldi Str. 1, 50411 Tartu, Estonia; meeri.visnapuu@ut.ee (M.V.); kristjan.saal@ut.ee (K.S.); dmytro.danilian@ut.ee (D.D.); rainer.parna@ut.ee (R.P.)

**Keywords:** antimicrobial, plywood lacquer, *Escherichia coli*, *Staphylococcus aureus*

## Abstract

In the context of healthcare-acquired infections, microbial cross-contamination and the spread of antibiotic resistance, additional passive measures to prevent pathogen carryover are urgently needed. Antimicrobial high-touch surfaces that kill microbes on contact or prevent their adhesion could be considered to mitigate the spread. Here, we demonstrate that photocatalytic nano-ZnO- and nano-ZnO/Ag-based antibacterial surfaces with efficacy of at least a 2.7-log reduction in *Escherichia coli* and *Staphylococcus aureus* viability in 2 h can be produced by simple measures using a commercial acrylic topcoat for wood surfaces. We characterize the surfaces taking into account cyclic wear and variable environmental conditions. The light-induced antibacterial and photocatalytic activities of the surfaces are enhanced by short-term cyclic wear, indicating their potential for prolonged effectivity in long-term use. As the produced surfaces are generally more effective at higher relative air humidity and silver-containing surfaces lost their contact-killing properties in dry conditions, it is important to critically evaluate the end-use conditions of materials and surfaces to be tested and select application-appropriate methods for their efficacy assessment.

## 1. Introduction

Bacterial infections are exerting a significant impact on public health. According to Global Health Metrics [1], microbial pathogens cause a higher burden to public health than either cancer or cardiovascular diseases. There is a clear link between bacterial infections and the healthcare environment. In Europe and its associated countries alone, healthcare-associated infections (HAIs) have been shown to affect 3.1–4 million people annually [2], out of which 90,000 may result in fatality [3]. An additional and increasing problem is the rapid spread of microbial resistance to existing treatments. According to the European Centre for Disease Prevention and Control (ECDC), antibiotic-resistant microbes are responsible for 33,000 annual deaths in Europe [4]. 

The most effective way to prevent the spread of microbial pathogens is to stop their transmission from person to person or via cross-contaminated surfaces. The latter is of significant importance as even up to 40% of microbial infections in hospital settings may spread through fomites, or microbe-contaminated materials [5,6]. Therefore, the introduction of antimicrobial surface materials that prevent the adhesion, proliferation or residence time of microbes would potentially provide great socioeconomic and health benefits [7]. In general, antimicrobial surfaces can be divided to those acting via the release of an antimicrobial agent, those acting upon direct contact between the microbe and the surface and to structured surfaces that usually avoid the binding of microbial cells to the surfaces [8]. The efficacy of antimicrobial coatings in reducing the microbial bioburden has been most clearly demonstrated in laboratory conditions but has also been studied in real-life situations, e.g., for copper surfaces [9,10,11], and their reducing activity on hospital-acquired infections has been shown [12]. Copper, along with silver, belongs to the most widespread class of antimicrobial surfaces, in the case of which the antibacterial active agent is constantly released from the coatings [13]. One of the drawbacks of such surfaces is the accumulation of dead bacterial mass on the surfaces and, thus, the blocking of the active and metal-ion-releasing surface [14]. This issue could be resolved by the use of reactive in-situ-acting photocatalytic surfaces. We have previously demonstrated that photocatalytically active TiO_2_ surfaces may, in addition to bacterial killing, also lead to the photooxidation of bacterial debris and, thus, the removal of dead bacterial mass from such surfaces, leading to the possibility for their extended use [15]. Besides pure photocatalytic materials, an encouraging approach in the context of more effective antimicrobial surfaces is the use of composite materials, which include a combination of different antimicrobial agents and mechanisms of antimicrobial action. Therefore, in our previous work [16,17], we developed composites of nanosized photocatalytic semiconductor material ZnO and metallic silver and proposed their antimicrobial use. The novelty of nano-ZnO/Ag composites arises from the combined and complex influence of different antimicrobial mechanisms: (i) surface-released or contact-acting Zn ions, (ii) surface-released or contact-acting Ag ions, (iii) light-induced reactive oxygen species (ROS) generated at the surface of the material. Moreover, due to the charge separation process in the ZnO/Ag system [18], the light-activated production of ROS is theoretically amplified compared with ZnO alone.

We have already demonstrated the antimicrobial efficacy of nano-ZnO/Ag composite materials both under UVA illumination and in dark conditions. Our previous studies have demonstrated 99.9% antibacterial activity of the surfaces in laboratory conditions under UVA that corresponded spectrally and in terms of intensity to natural solar light [17]. However, these previous experiments concerned a layer of bare nano-ZnO/Ag particles heat-annealed to glass surfaces, where no additional substances (matrix) were used to fix the nanomaterial to the surfaces. The tight surface attachment of the antimicrobial components would be necessary for the prolonged use of touch surfaces. Similar issues related to claiming antimicrobial activity solely based on the efficacy testing of antimicrobial components and not based on the efficacy of the final end-products are true for many research papers but also commercial products. Therefore, in this paper, we aim at demonstrating the antimicrobial activity of nano-ZnO/Ag materials whose antibacterial efficacy has been previously tested, in their application-relevant configuration. In the paper, we demonstrate the process of embedding antimicrobial ZnO/Ag nanomaterials into an acrylic-based commercial topcoat matrix that can easily be used in industry without changing the existing technology and causing excess costs. The matrix with embedded nano-ZnO and nano-ZnO/Ag was applied to plywood, which is one of the materials to be covered with the acrylic-based matrix for various real-life applications. Therefore, this study moves beyond traditional laboratory testing, where, usually, glass is selected as a standard substrate material. To further increase the real-life relevancy of our coatings, we evaluated the performance of matrix-embedded nano-ZnO and nano-ZnO/Ag over several use cycles, with organic soiling as well as at differing air humidity levels. Throughout the paper, we discuss the contribution of photocatalytic activity and release of antibacterial ions of Zn^2+^ and Ag^+^ to the antibacterial action of our surfaces. Altogether, our results are expected to pave the way towards the use of nano-ZnO- and nano-ZnO/Ag-based surface coatings in fit-for-purpose antimicrobial surface applications.

## 2. Materials and Methods

### 2.1. Synthesis of Nano-ZnO and Nano-ZnO/Ag Particles and Their Characterization

Nano-ZnO and nano-ZnO/Ag particles were prepared similarly to our previous works [16,17]. Briefly, zinc acetate was used as a precursor in an alkaline environment under reflux conditions to synthesize nano-ZnO. Nano-ZnO was supplemented with Ag using UVA-induced photodeposition to prepare nano-ZnO/Ag composite particles. XRD analysis of ZnO/Ag composite particles showed the presence of crystalline ZnO with a wurtzite structure (ICDD PDF2011, 01-070-8072).

Scanning transmission electron microscopy–energy-dispersive X-ray spectroscopy (STEM–EDX) analysis was used to characterize the morphology and elemental content of nano-ZnO/Ag. First, 200 μg/mL particle suspension was deposited onto a 400-mesh holey carbon-coated copper grid (Agar Scientific Ltd, Stansted, UK) and analyzed using a Titan Themis 200 Cs corrected transmission electron microscope (TEM, Thermo Fisher Scientific Inc. (formerly FEI), Waltham, MA, USA) at 200 kV. EDX signal of the particles was collected with the SuperX silicon drift detector (Bruker, Billerica, MA, USA) in STEM mode. The high-angle annular dark-field (HAADF) STEM image was combined with EDX element mapping and signals for silver L_α_, zinc K_α_ and oxygen K_α_ were extracted to visualize their elemental distribution.

The concentration of Zn and Ag in nano-ZnO and nano-ZnO/Ag suspensions used for preparing nanoparticle-covered substrates and embedded nanoparticle-based coatings was analyzed using total reflection X-ray fluorescence (TXRF; S2 Picofox, Bruker, Billerica, MA, USA) for Zn and an atom absorption spectrometer (AAS; ContrAA 800, Analytik Jena AG, Jena, Germany) for Ag, followed by acidification of nanoparticle suspensions with 1% HNO_3_.

### 2.2. Preparation of Surfaces Where Nano-ZnO and Nano-ZnO/Ag Are Embedded in Acrylic Matrix

First, 25 × 25 mm squares of plywood (birch, 4 mm) were used as substrates to be covered with acrylic-embedded nano-ZnO and nano-ZnO/Ag. 

Acrylic clear matt commercial two-component topcoat meant for wooden surfaces (topcoat TZ9310/00 and hardener TH0790/00, Sayerlack, Pianoro, Italy) was used as the acrylic matrix to embed the prepared nano-ZnO and nano-ZnO/Ag particles. To mix nano-ZnO or nano-ZnO/Ag to TZ9310/00, the particles were first dispersed in the solvent (thinner DT0013/00, Sayerlack, Pianoro, Italy) used in the topcoat mixture and then mixed with the rest of the topcoat components. 

Prior to covering the wooden surfaces with nanomaterial-supplemented acrylic matrix, the surfaces were undercoated (TU0141/00, Sayerlack, Pianoro, Italy) and left to dry in ambient laboratory conditions for a week. Nano-ZnO and nano-ZnO/Ag supplemented topcoat was then evenly applied onto the surfaces using a brush. Brushing was used to mimic a possible real-life approach for applying the coating. Hence, the thickness of the coating was a result of the chosen coating technique and was not further optimized. The surfaces were left to dry at ambient laboratory conditions for a week before subsequent analysis. Wooden surfaces covered with undercoat and topcoat without any nanomaterial addition were used as negative controls.

For “nanomaterial controls” in antimicrobial and photocatalytic experiments, glass surfaces with (matrix-free) nano-ZnO and nano-ZnO/Ag particles were used. For this, nano-ZnO and nano-ZnO/Ag suspensions were spin-coated onto 25 × 25 mm square cover glasses (2855-25, Corning Inc., Corning, NY, USA). The surfaces were subsequently heated at 200 °C for 6 h (similarly as described in our former study [17]) to improve nanoparticle adherence to the glass substrates and ensure the removal of organic residues, both of which can specifically affect the outcome of antimicrobial activity study (e.g., possible toxic effects of organic residues and attachment of loose particles onto bacterial cells). As we did not observe differences in the photocatalytic activity of annealed and unannealed nanoparticle-covered glass surfaces, we decided to proceed with annealed samples to reduce the possibility of the mentioned side effects in the antimicrobial study. To analyze the content of Zn and Ag in these surfaces, the coverslips were treated with 1 mL concentrated HNO_3_ for 1 h, after which the solution was diluted with water and analyzed using TXRF or AAS. Bare cover glasses were used as controls in these experiments.

### 2.3. Characterization of Matrix-Embedded Nano-ZnO- and Nano-ZnO/Ag-Based Surfaces

#### 2.3.1. Visualization of the Surface Coatings Using Electron Microscopy

For scanning electron microscopy (SEM) analysis, the upper layer of matrix-embedded nano-ZnO- and nano-ZnO/Ag-covered plywood surfaces was removed and attached onto a 45-degree angled SEM pin stub using conductive silver paint. Top view and cross-section of the initial samples and samples after 5 and 10 reuse cycles were imaged by Nova NanoSEM 450 (Thermo Fisher Scientific Inc. (formerly FEI), Waltham, MA, USA) at 15 kV using a low vacuum detector at 100 Pa.

#### 2.3.2. Elemental Composition and Chemical State of Elements on the Surfaces

Chemical state and electronic structure of the matrix-embedded nano-ZnO and nano-Ag/ZnO on wood surfaces were investigated with X-ray photoelectron spectroscopy (XPS). Samples were first degassed in vacuum, until the base pressure was 1 × 10^−8^ mbar, and then heated twice to 60 °C for 1 hour. XPS measurements were conducted by using an electron energy analyzer, the Scienta SES-100, and a non-monochromatized twin-anode X-ray source, the Thermo XR3E2 (Mg K_α_ 1253.6 eV and Al K_α_ 1486.6 eV). The Ar^+^ ion sputtering was performed in ultra-high vacuum conditions at 10^−8^ mbar. The duration of the Ar^+^ beam exposure was 15 min and the Ar^+^ current was 10 mA. The spot size was adjusted to cover the sample. The binding energy scales for the XPS experiments were referenced to the binding energy of C 1s (284.8 eV) photoemission line. Data analysis was carried out by using CasaXPS software (version 2.3.12, Casa Software Ltd., Teignmouth, UK) [19]. Zn 2p, Ag 3d, O 1s, C 1s, Cl 2p and Si 2p photopeaks were fitted by using symmetric Gaussian–Lorentzian line shapes after subtracting a Shirley-type background.

#### 2.3.3. Analysis of Zn Content on Surfaces

Wavelength-dispersive X-ray fluorescence (WD-XRF) spectroscopy was used to determine the Zn concentration in matrix-embedded nano-ZnO- and nano-ZnO/Ag-covered plywood surfaces. The WD-XRF 4 kW instrument (ZSX Primus II, Rigaku, Tokyo, Japan) was calibrated using LabKings SM68 Standard 1 as a reference. The set amount of standard was pipetted onto control coatings (pure topcoats without added nanoparticles) and the solvent was evaporated by heating at 70 °C in an exicator. Calibration curve was established using 5 reference concentrations.

#### 2.3.4. Release of Zn and Ag from the Surfaces

Release of Zn and Ag from matrix-embedded nano-ZnO- and nano-ZnO/Ag-covered surfaces was measured in conditions comparable to antimicrobial tests. First, 20 μL of test medium was pipetted onto the surfaces and incubated under 20 mm × 20 mm × 0.05 mm polyethylene (PE) film for a specified duration. The liquid was then washed off with 2 mL deionized water. Zn content in the wash-off was measured by TXRF (S2 Picofox, Bruker, Billerica, MA, USA) and Ag content by AAS (ContrAA 800, Analytik Jena AG, Jena, Germany).

#### 2.3.5. Contact Angle Measurements

The sessile drop method [20] in combination with a moving platform (based on Thorlab DT12 dovetail translation stage) was used to measure the contact angles of matrix-embedded nano-ZnO- and nano-ZnO/Ag-covered wood surfaces. A drop of deionized water (2 μL) was pipetted onto a dry surface. After 10 seconds, the water droplet was photographed with a Canon EOS 650d camera using a MP-E 65 mm f/2.8 1–5× macro focus lens. The contact angle formed by the liquid drop on the wood surface was determined using image analysis software (ImageJ 1.8.0 172 for Windows, plugin for contact angle measurement [21], National Institutes of Health, Bethesda, Maryland, USA, plugin by Marco Brugnara (marco.brugnara at ing.unitn.it)). Contact angles were measured from 4 water droplets per sample, each droplet pipetted at a different location.

### 2.4. Evaluation of Antibacterial Efficacy of Matrix-Embedded Nano-ZnO- and Nano-ZnO/Ag-Based Surfaces

Gram-positive and Gram-negative model organisms routinely used in standard protocols were selected for the assessment of antibacterial efficacy. *Escherichia coli* DSM1576 (ATCC8739) was ordered from the German Collection of Microorganisms and Cell Cultures (DSMZ, Braunschweig, Germany) and *Staphylococcus aureus* ATCC6538 was acquired from the National Institute of Chemical Physics and Biophysics (Tallinn, Estonia). For bacterial suspensions used in the tests described below, bacteria were cultivated overnight on nutrient agar plates (NA: 5 g/L meat extract, 10 g/L peptone, 5 g/L NaCl, 15 g/L agar), collected by using a sterile inoculation loop and suspended in exposure medium (500-fold diluted nutrient broth: 3 g/L meat extract, 10 g/L peptone, 5 g/L NaCl in undiluted broth) followed by adjusting cell density by means of optical density measurement and further dilution in exposure medium based on prior calibration. Bacteria were harvested from the surfaces by submerging the surfaces in 15 mL of toxicity-neutralizing soybean casein digest broth with lecithin and polyoxyethylene sorbitan monooleate (SCDLP: 17 g/L casein peptone, 3 g/L soybean peptone, 5 g/L NaCl, 2.5 g/L Na_2_HPO_4_, 2.5 g/L glucose, 1.0 g/L lecithin, 7.0 g/L nonionic surfactant Tween80) in 50 mL centrifugation tubes, followed by vortexing the tubes for 30 s at full power, serial dilutions in phosphate-buffered saline (PBS: 8 g/L NaCl, 0.2 g/L KCl, 1.63 g/L Na_2_HPO_4_ × 12H_2_O, 0.2 g/L KH_2_PO_4_; pH 7.1), drop-plating on NA plates and incubation at 30 °C. Colonies of *E. coli* were counted 24 h and *S. aureus* 48 h after plating. Results are expressed as log-transformed CFU counts. All aqueous media were prepared and/or diluted in sterile deionized water and autoclaved to sterilize. Bovine serum albumin (BSA, Merck KGaA, Darmstadt, Germany) was filter-sterilized and added to soil load solution after autoclaving and cooling to room temperature. 

#### 2.4.1. Surface Antibacterial Testing Using ISO 22196 and ISO 27447 Methods

Antibacterial testing was carried out using a test protocol modified from ISO 22196 and ISO 27447 standard procedures adapted to the studied materials and the need for higher throughput due to several materials and time points studied, as described by Visnapuu et al. [17]. Precultured bacteria were suspended in 500-fold diluted NB to achieve an OD600 value of 0.074 for *E. coli* and 0.070 for *S. aureus*, resulting in approximately 3.5 × 10^5^ CFU/cm^2^. Then, 20 μL of the prepared inoculum suspension was applied to 25 × 25 mm sample surfaces, covered by 20 × 20 × 0.05 mm PE film and incubated in the dark or exposed to UVA in parallel. PE cover film was used to attain even thin coverage of bacterial suspension with good contact between microbes and the surface and even UVA exposure, as well as to ensure similar humidity conditions on the predefined test area. Exposures were carried out in a humid environment on a bent glass U-rod over sterile wet filter paper covered by a Petri dish in dark conditions or 1.1 mm thick UVA-transmissive borosilicate moisture preservation glass under UVA illumination. Temperature (22 °C) and lighting (2.43 ± 0.11 W/m^2^ at 315–400 nm spectral range, measured using Delta Ohm UVA probe) were achieved in a climate chamber, the MMM Climacell 111 EVO, equipped with a UVA/Vis Combi light tray (MMM Medcenter Einrichtungen GmbH, Planegg, Germany). Exposures were terminated, cells harvested and viable counts determined as described above.

#### 2.4.2. Analysis of the Effect of Relative Humidity on Antibacterial Efficacy

Bacterial suspensions were prepared as described above, but applied to the surfaces in 1 μL droplets in the same volume and final cell counts per surface. The surfaces were incubated under the same lighting conditions as above but left uncovered and kept at different relative air humidities. Only 1 h time point was collected at each humidity condition, temperature (22 °C) and lighting (2.43 ± 0.11 W/m^2^ at 315–400 nm spectral range achieved in climate chamber MMM Climacell 111 EVO as specified above). Exposures were terminated, cells harvested and viable counts determined as described above.

#### 2.4.3. Analysis of Residual Activity of Used Surfaces

To analyze the continuing antimicrobial effect of matrix-embedded nano-ZnO- and nano-ZnO/Ag-based surfaces, the surfaces were subjected to 10 rounds of soiling with a 10 μL organic soil load [22] (2.5 g/L BSA, 3.5 g/L yeast extract, 0.8 g/L mucin in PBS), 30 min incubation at 22 °C, 50% RH, 20 W/m^2^ UVA and wiping with a moist, sterile microfiber cloth after each round. Virgin surfaces and surfaces subjected to 5 or 10 rounds of reuse were tested for residual antibacterial activity using the modified ISO procedure as described above. Only *E. coli* as the more sensitive organism was used to evaluate the effect of wear on antibacterial activity.

#### 2.4.4. Determination of Minimal Biocidal Concentrations of Zn and Ag Ions and Peroxide

Minimal biocidal concentration (MBC) of ZnSO_4_, AgNO_3_ and H_2_O_2_ at 1, 2 and 4 h towards *E. coli* and *S. aureus* was determined in 1:500 NB at room temperature to evaluate sensitivity of the bacterial strains used to respective metal ions and ROS damage. For the analysis, in a 96-well plate, 100 μL of OD600 = 0.2 bacterial suspension in 1:250 NB prepared from overnight plate culture was combined 1:1 with 100 μL of 2× dilution series of substance of interest in deionized water. Plate was incubated in dark ambient conditions in the biosafety cabinet. At 1, 2 and 4 h, a 3 μL drop was plated on NA from each well. MBC was determined 48 h post-plating as the lowest substance concentration completely preventing growth after exposure.

### 2.5. Photocatalytic Activity of Matrix-Embedded Nano-ZnO- and Nano-ZnO/Ag-Based Surfaces

The degradation of methylene blue (MB) as a model for an organic contaminant was used to determine the level of photocatalytic activity. First, 2 mL of 1.6 × 10^−5^ M MB aqueous solution was pipetted onto the prepared surfaces and exposed to UVA irradiation or kept in dark conditions for control in a climate chamber (22 °C, 90% rh). The pH of the initial MB solution was 5.3, which is close to the recommended pH value (pH 5.5) for the solution [23]. After predetermined irradiation times, the dye solution was pipetted from the surfaces into a standard PS 10 mm cuvette to measure the UV–Vis absorbance of MB using a spectrophotometer (Cary UV-Vis-NIR 5000, Agilent Technologies, Inc., Santa Clara, CA, USA). Virgin surfaces were tested in 3 parallels under UVA irradiation and in dark conditions for 15 min to 12 h, and surfaces subjected to 5 or 10 rounds of reuse (described above) were tested in 3 parallels under UVA irradiation for 4 h. The absorbance intensity at 663 nm (characteristic absorbance peak for MB) after exposure was compared to the initial intensity to evaluate the degradation of MB. Before exposure, the surfaces with dye solution were pre-conditioned in the climatic chamber in dark conditions for 20 min to establish the adsorption equilibrium of MB. A self-built lamp consisting of 4 fluorescent Hg light bulbs (15 W iSOLde Cleo, λ_max_ = 355 nm) was used and the light intensity at test surface height was 2.8–3.2 W/m^2^ at 315–400 nm spectral range (measured using Delta Ohm UVA probe).

### 2.6. Statistical Analysis

One-way or two-way ANOVA analysis followed by Tukey post-hoc test was used to detect significant differences in multiple comparisons at 0.05 significance level using GraphPad Prism 9.1.2 (GraphPad Software, La Jolla, CA, USA). Log-transformed data at or above the detection limit of 3 colonies counted in undiluted plated drop were used for the analysis of CFU counts. Only statistically significant (*p* < 0.05) differences and log reductions are mentioned in the text, unless stated otherwise.

## 3. Results

### 3.1. Characterization of Matrix-Embedded Nano-ZnO- and Nano-ZnO/Ag-Coated Surfaces

The synthesized nanoparticles of ZnO/Ag composites are shown in Figure 1. STEM–EDX imaging was used to analyze the morphology and elemental content of nano-ZnO/Ag. The upper left image in Figure 1 demonstrates a HAADF-STEM image of ZnO/Ag nanoparticles. HAADF-STEM images combined with EDX mapping demonstrate the distribution of silver (Ag, L_α_) in blue, zinc (Zn, K_α_) in violet and oxygen (O, K_α_) in red. The HAADF-STEM images of ZnO nanoparticles (not shown on figure) are similar, but without small Ag-based nanoparticles. The homogeneous distribution of elemental Zn and O over the rod-shaped nanoparticles confirms that these nanoparticles are based on ZnO. The distribution of Ag demonstrates that silver nanoparticles are much smaller than ZnO nanorods and are located on the surfaces of larger ZnO nanoparticles, as expected. 

Elemental analysis indicated that, in the suspensions of nano-ZnO and nano-ZnO/Ag that were used to prepare the nano-ZnO and nano-ZnO/Ag acrylic matrices, the Zn concentration was 50,000 µg/mL (6.2 wt% ZnO), and in the nano-ZnO/Ag suspension, the Ag concentration was 350 µg/mL (0.4 mol% ZnO).

Figure 2 demonstrates photographic (first row) and SEM images of the coatings on wooden substrates. The top view (second row) and cross-section (third row) SEM images reveal a similar distribution for both nano-ZnO- and nano-ZnO/Ag-embedded nanoparticles. Cross-sections demonstrate that the distribution of embedded nanoparticles at nanometer scale is not completely homogeneous. However, it is clear that, since the acrylic matrix material precursor is relatively viscous already before drying, it is unrealistic to expect a completely homogeneous distribution of nanoparticles. The lowest row of Figure 2 also demonstrates the Zn content (µg/cm^2^) determined using XRF. The Zn content in both nanoparticle-based coatings was similar and much higher compared to the control (pure matrix-covered substrate). It is important to note that XRF provides information about the elemental content of bulk material, i.e., in the case of our samples, XRF collected information from the whole coating thickness, along with some signals from the plywood substrate.

XPS was used to characterize the surface regions of the samples. Differently from XRF, X-ray photoelectron microscopy is a surface-sensitive method and it collects data only from the topmost parts (some nm) of the surface. Since microbe/surface interaction and also dissolution processes take place at the surface region, XPS data are very informative in this context. From the overview spectra of nano-ZnO- and nano-ZnO/Ag-containing coatings on plywood substrates (Figure 3a), the presence of oxygen, carbon and zinc can be seen in the surface region. Moreover, weak peaks associated with chlorine, silicon and nitrogen were observed. The overview spectrum of nano-ZnO/Ag-containing coatings does not show a characteristic silver 3d photoline since the amount of silver is below the XPS detection level of our set-up (roughly 0.1 atomic percent). Figure 3b demonstrates Zn 2p high-resolution XPS spectra of the same samples. According to the Zn 2p line positions and shape, it can be concluded that Zn is in the Zn^2+^ oxidation state, as expected in the case of ZnO. More detailed XPS spectra for the same samples are shown in Appendix A (O 1s, N 1s, C 1s, Cl 2p and Si 2p). Appendix A demonstrates also a region corresponding to the Ag 3d location in the XPS spectra of the nano-ZnO/Ag-containing coating on the wood substrate before and after Ar^+^ sputtering, i.e., “as prepared” and after the removal of the topmost layer of sample with Ar^+^ ion bombardment. No Ag signal was detected in either case, likely due to the insufficient sensitivity of XPS to the low levels of Ag present in the ZnO/Ag composites.

Table 1 demonstrates the atomic concentrations of elements in nano-ZnO- and nano-ZnO/Ag-containing coatings on wood substrates according to XPS analysis. For comparison, similar data for pure matrix material on wood substrate are given. XPS of nano-ZnO/Ag does not show Ag photolines, due to the insufficient sensitivity of the XPS, as discussed above. 

### 3.2. Antibacterial Activity of Matrix-Embedded Nano-ZnO- and Nano-ZnO/Ag-Based Surfaces

In order to assess the antibacterial activity of the matrix-embedded nano-enabled surfaces, antibacterial testing was performed following an in-house modification of ISO 22196 and ISO 27447 standard methods for higher throughput due to several materials and time points tested. The mentioned methods are based on a thin layer of liquid bacterial inoculum in contact with a large surface area [24] and are routinely used for solid and/or photocatalytically active antimicrobial surface testing, representing a common test format required for verifying the antimicrobial claims of respective commercial products [25]. 

Both of the matrix-embedded nano-enabled surfaces demonstrated the maximum quantifiable antibacterial effect towards both bacteria after 2 h exposure under low-intensity UVA illumination, with >2.8 logs (*p* < 0.0001) and >2.7 logs (*p* < 0.0001) of reduction in the viability of *E. coli* and *S. aureus*, respectively (Figure 4a,b). These values are approaching the minimum 3-log reduction requirements in up to 2 hours to apply for public health claims in the US [26] and EU [25]. Interestingly, the addition of silver to the ZnO NPs seemed to enhance the photocatalytic efficiency of the matrix-embedded surfaces and not bare nano-covered surfaces (Figure 4e). However, this additional effect of silver on matrix-embedded nano-ZnO/Ag was not strong enough to be clearly reflected in the antibacterial activity, possibly due to differences in the test formats for organic dye degradation and ISO-based antimicrobial testing, as well as differences in the sensitivities of the tests as dye degradation is, in essence, a chemical reaction, whereas bacteria are evolutionarily adapted to actively counteract ROS attack [27].

The antibacterial activity of the matrix-embedded nano-ZnO/Ag surfaces was similar, with a statistically significant but biologically moderate effect towards both *E. coli* and *S. aureus* after 4 h of dark exposure, with 1.2 (*p* = 0.0009) and 1.1 (*p* = 0.0002) log reductions in viability, respectively (Figure 4a,b). However, the nano-enabled surfaces seemed to act quicker on *E. coli*, followed by plateauing, with an average of 1.5 (*p* = 0.0007) and 1.2 (*p* = 0.0009) log reductions on nano-ZnO/Ag compared to the control at 1 and 4 h, respectively, while the corresponding reduction values for *S. aureus* were non-significant, at 0.1 logs (*p* = 0.9985) after 1 h and 1.1 logs (*p* = 0.0002) after 4 h of dark exposure. No statistically significant antibacterial activity of matrix-embedded nano-ZnO towards *S. aureus* after 4 h dark exposure was detected and its activity towards *E. coli* was biologically negligible (0.8 logs; *p* = 0.04).

As stated above, we have previously tested the antimicrobial activity of solid surfaces based on nano-ZnO and nano-ZnO/Ag heat-annealed to glass carriers in different coverage densities [16,17] and showed substantial antibacterial activity already after 15–30 min under UVA illumination. These surfaces with “bare” nano-ZnO and nano-ZnO/Ag (Appendix A) were also used in this study as a comparison. We demonstrated that while a significant decrease in viable bacterial counts on matrix-embedded nano-ZnO surfaces was obtained only after 60 minutes (Figure 4c,d), and an even longer time, 120 minutes, was required to achieve the maximum detectable decrease in bacterial viability on matrix-embedded nano-ZnO and nano-ZnO/Ag (Figure 4a–d), on bare nano-ZnO surfaces, an analogous antibacterial effect was obtained already after 15 minutes (Figure 4c,d). In accordance with the antibacterial results, the abiotic photocatalytic effect of the bare nano-enabled surfaces was higher than that of the matrix-embedded nano-ZnO and nano-ZnO/Ag surfaces. The lower efficacy of the matrix-embedded nanomaterials was expected and could be explained by the fact that less of the active surface of nano-ZnO and nano-ZnO/Ag was exposed to the bacterial cells and dye (nanoparticle surface was partly physically blocked by the matrix material), reducing direct contact between the cells or dye and nanoparticles. Interestingly, under UVA illumination, the bare nano-ZnO/Ag caused 1.2 logs more reduction in *E. coli* and *S. aureus* viability (*p* = 0.0018 and *p* < 0.0001, respectively) than matrix-embedded nano-ZnO/Ag. Respective statistically significant differences between these two bacterial species on nano-ZnO-enabled surfaces could not be determined. There were no statistically significant differences between the matrix-embedded NP surfaces and bare NP surfaces in dark conditions. 

### 3.3. Antibacterial Activity of Surface-Released Active Agents

To control for the possibility of antibacterial effects to be caused by biologically active metals released from the surfaces of the coatings, the amount of released Zn and Ag was measured from antibacterial test set-up in the ISO test format (Figure 5a,b). To understand the potential antibacterial effect of surface-released Zn and Ag, the sensitivity of the test bacteria in the form of minimal biocidal concentration (MBC) towards respective metal salts was determined in the same timeframe and medium used in the testing of the surfaces (MBC; Figure 5c–e). The sensitivity of the bacteria towards hydrogen peroxide was used as an indicator of their susceptibility to ROS. The mean concentration of Ag released from the surface coatings in our test settings was below the detection limit and therefore also below the MBC value. Based on previous knowledge [17], we can assume that the release of Ag is higher in dark conditions, while more Zn is released under UVA illumination, and this photo-induced process seems to be further enhanced in the presence of Ag (Figure 5a,b). Noble metal deposits on semiconductor particle surfaces can act as electron sinks, facilitating charge separation and improving photocatalytic efficiency. Due to the more effective charge separation, different photoinduced chemical processes take place more effectively, and this may be the reason that Ag deposits enhance Zn^2+^ release under UVA illumination. Mean concentrations of Zn in the test system did not reach MBC values, suggesting that any toxic effects of the surfaces in dark conditions could not be caused by ionic toxicity but rather surface contact or the combined toxic effects of the multimodal surfaces. Based on the MBC values, *E. coli* was clearly more sensitive to ROS toxicity caused by hydrogen peroxide (Figure 5e), while *S. aureus* was slightly more sensitive to zinc (Figure 5c). Sensitivity to silver could be considered similar (Figure 5d).

### 3.4. Antibacterial Activity in Variable Relative Humidity

It is important to note that methods based on the original Japanese industrial standard JIS Z 2801 and its derivatives, including the ISO standards 22196 and 27447, offer a robust protocol for the evaluation of the antibacterial properties of solid surfaces in laboratory conditions but pose critical limitations in terms of extrapolating the results to real-life applications of the tested materials. These tests are useful for the basic concept of comparing materials during development but not for assessing the antibacterial properties of the end-product. The antimicrobial efficacy of high-touch surfaces in particular could be overestimated during long exposures in the warm and moist “best-case scenario” of the mentioned standard protocols as opposed to drier and cooler ambient conditions in most end-use scenarios. Relative humidity has been shown to dramatically affect bacterial viability [28] and antimicrobial surface efficiency [29] and can be challenging to include in a test format [30]. Thus, in parallel to the test conditions resembling ISO standard conditions, the antimicrobial activity of the matrix-embedded nano-ZnO- and nano-ZnO/Ag-based surfaces was also tested by placing several 1 μL droplets on the tested surfaces, followed by a relatively short 1-hour incubation period in different relative air humidity (RH) conditions at room temperature. RH values were selected to resemble the extremely dry indoor conditions encountered during the heating season (15% RH), normal ambient RH range (50% RH) and almost saturated conditions commonly encountered during antibacterial testing (90% RH) in comparison with wet conditions in the ISO testing format (>90%) at the same temperature (22 °C). It is clearly evident from Figure 6 that the nano-enabled surfaces generally demonstrate higher antibacterial ability at higher RH. In the context of silver being the most used active ingredient in antimicrobial surface applications [31], it is especially noteworthy that silver-based surfaces lose antibacterial activity towards *E. coli* in dry dark conditions (Figure 6a). The viability of *E. coli* on neither nano-ZnO nor nano-ZnO/Ag surfaces is significantly different from the control surface in otherwise same conditions at 15% and 50% RH, while a 1.5 log reduction (*p* = 0.0007) in *E. coli* viability compared to the control was registered in ISO conditions. Due to the fact that the mean Ag concentration in the system for the modified ISO test system was below the bactericidal values (Figure 5d), we propose that the silver-based surfaces lose their contact-killing properties in dry conditions. The same has been demonstrated by Michels et al. towards methicillin-resistant *Staphylococcus aureus* [29] and Knobloch et al. towards *Enterococcus faecium* [32]. In our case, the silver content in the nano-enabled surfaces was most probably too low to elicit antibacterial activity towards *S. aureus* (Figure 6b). The photo-induced antibacterial activity of nano-ZnO towards *E. coli* (1–2 logs decrease in viability) does not seem to be affected by RH (not quantifiable at 15% RH). Towards *S. aureus*, the photo-induced antibacterial activity of nano-ZnO does not seem to follow a linear change and demonstrates the highest activity at the lowest RH and in wet ISO conditions but not at moderate RH, with 1.7 (*p* < 0.0001) and 1.9 (*p* < 0.0001) log reductions in viability, respectively. Apart from the effect of active agents in the surfaces, it is evident that low-intensity UVA exposure is more phototoxic to both bacteria in dry conditions (Figure 6a,b) and the Gram-negative *E. coli* is, as expected, more sensitive to drying (Figure 6a) than the Gram-positive *S. aureus*. The combination of the two environmental variables renders it impossible to compare the antibacterial activity of the materials in UVA-exposed dry conditions towards *E. coli* (Figure 6a) decreasing viability on the control surface to the limit of quantification. 

Differences in antibacterial activity caused by the different test conditions in standard protocols versus more application-appropriate conditions of variable RH clearly indicate that antibacterial activity is critically dependent on the end-use conditions as well as the selection of target species. Therefore, the use of multimodal antimicrobial surfaces combining several mechanisms of action could result in robust results in terms of efficiency.

### 3.5. Residual Activity of Nano-ZnO and Nano-ZnO/Ag Surfaces

Antibacterial surfaces are often evaluated only for their antibacterial activity and not for the effect of cleaning or wearing on this activity [31]. To assess the effect of the reuse of the surfaces on their antibacterial activity, we subjected the surfaces to a cyclic wear regimen with organic soiling, high-intensity UVA exposure and wiping with a sterile, moist microfiber cloth. Antibacterial activity towards *E. coli* (Figure 7a), photocatalytic activity (Figure 7b) and the water contact angle (Figure 7c) of virgin surfaces as well as after 5 and 10 reuse cycles were measured as above. No statistically significant changes in the antibacterial activity of the virgin nano-ZnO surfaces or those after 5 or 10 reuse cycles were detected in dark conditions. The virgin nano-ZnO/Ag surfaces demonstrated a 1.0 log greater decrease in bacterial viability than after 10 reuse cycles (*p* = 0.006), with respective antibacterial activities of 1.5 logs (*p* < 0.0001) reducing to a non-significant 0.5 logs. However, the antibacterial activity of the nano-enabled surfaces in UVA-irradiated conditions increased after 10 reuse cycles, with at least a 1.0 log reduction in bacterial viability for the nano-ZnO (from 2.4 to above 3.4; *p* = 0.0032) and a moderate 0.9 log difference for nano-ZnO/Ag (from 1.9 to 2.8; *p* = 0.02). While organic dye degradation measurements showed variable results after five reuse cycles, a statistically significant increase in photocatalytic activity was detected for both nano-ZnO (18%; *p* = 0.01) and nano-ZnO/Ag surfaces (22%; *p* = 0.003). The nano-ZnO/Ag surfaces were more active than the nano-ZnO surfaces in the case of the virgin surfaces (15%; *p* = 0.03) and after 5 (40%; *p* < 0.0001) and 10 (19%; *p* = 0.007) reuse cycles. The small decrease (less than 3% in most cases) in MB UV–Vis absorbance intensity in the case of pure matrix-covered wooden surfaces can be associated with MB adsorbance onto the coating and not with dye degradation (Figure 7b). A decrease in surface hydrophobicity and better wetting with soil load was empirically observed after 3–5 cycles of reuse. This is reflected in the decreased and less variable water contact angles after 10 reuse cycles (Figure 7c), with an approximately 10°–13° decrease on all surfaces including control surfaces. This indicates that increased hydrophilicity upon reuse is a property of the matrix material and not the added nanoparticles.

We have previously demonstrated that the antibacterial activity of surfaces covered with bare nanoparticles decreases with reuse cycles even without physical wiping [17]. Loss of antibacterial activity on the matrix-embedded nano-ZnO/Ag surfaces after 10 reuse cycles in dark exposure conditions does indicate a depletion or masking of the surface-exposed active ingredient. Contrary to what was expected, an increase in the antimicrobial activity of the matrix-embedded nano-enabled surfaces under UVA irradiation was demonstrated, indicating the better stability and functionality of the matrix-embedded nano-enabled surfaces. As the nano-ZnO surfaces demonstrated higher antibacterial activity than nano-ZnO/Ag while the latter had better photocatalytic properties, we suggest that it is the combination of the photocatalytic effect, changes in Zn release profiles and Ag photoreduction possibly masking some of the ZnO active surface that causes the antibacterial properties in UVA-irradiated conditions, as more Zn was released from the virgin nano-ZnO/Ag surfaces than from the nano-ZnO surfaces (Figure 5a,b). The fact that the antibacterial, photocatalytic and hydrophilic properties of the nano-enabled surfaces increased after 10 reuse cycles indicates that it could be due to the cyclic high-intensity UVA irradiation, resulting in surface damage and exposing more of the active surface of the nanoparticles. However, the extent of possible damage was not visually detectable on the SEM images of virgin and reused surfaces (Figure 8).

## 4. Conclusions

In the present work, we prepared and analyzed the properties of antimicrobial coatings based on ZnO and ZnO/Ag nanoparticles embedded into a commercially available, acrylic clear matt topcoat meant for wooden surfaces and exhibiting good water resistance and mechanical, chemical and visual stability. We were successful in evenly distributing the nanoparticles of ZnO and ZnO/Ag in the acrylic matrix. XPS data showed that a remarkable amount of ZnO nanoparticles was located in the topmost region of the surface coating (some nm) and, thus, they were able to exhibit antibacterial effects. However, the measured antibacterial effect was significantly slower than that of bare nanoparticles. The surfaces based on matrix-embedded nano-ZnO and nano-ZnO/Ag demonstrated substantial antibacterial activity under UVA illumination in standard wet conditions, with at least a 2.7-log reduction in bacterial viability in 2 h. The addition of Ag to the surface coating matrix granted moderate antibacterial activity also in dark conditions after 4 h. Ag also enhanced photocatalysis and Zn release from the matrix-embedded surfaces under UVA illumination but did not affect the antibacterial properties under UVA. Neither metal release from the surfaces nor photocatalytic activity could explain the antibacterial effect in all conditions, confirming the multimodal antibacterial action of the surfaces resulting from photocatalysis, Zn and Ag ion release and contact killing. Antimicrobial efficacy testing at variable application-relevant relative air humidity revealed that better antibacterial efficacy was achieved at higher humidity and silver lost its contact-killing properties in dry conditions, highlighting the need to use a methodology that best reflects the end-use conditions. The UVA-induced antibacterial and photocatalytic activities of the surfaces were enhanced by short-term cyclic wear, indicating the potential for prolonged effectivity in real use situations.

As a future outlook, antimicrobial surfaces with sufficient efficacy have certainly huge potential in maintaining good hygiene and a healthy environment, especially in vulnerable settings and public spaces. However, such surfaces should be developed in a sustainable manner, considering the potential environmental effects, including the long-term durability of the surfaces, as well as the minimization of the risk of the development of antimicrobial resistance.

## Figures and Tables

**Figure 1 nanomaterials-11-03384-f001:**
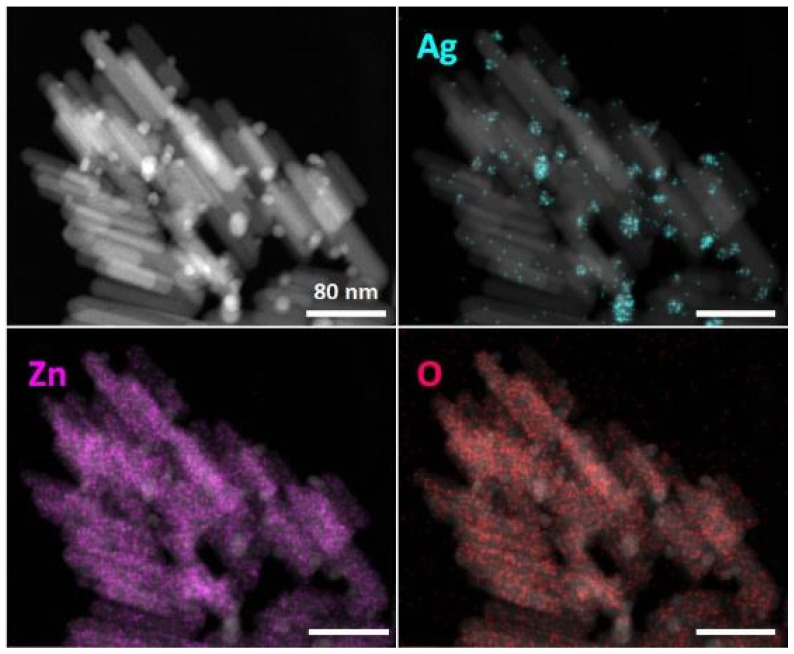
STEM–EDX analysis of ZnO/Ag nanoparticles. HAADF-STEM images combined with EDX mapping results for silver (Ag, L_α_) in blue, zinc (Zn, K_α_) in violet and oxygen (O, K_α_) in red show the homogeneous distribution of elemental Zn and O and the presence of Ag in nanoparticle formulations. Scale bars correspond to 80 nm.

**Figure 2 nanomaterials-11-03384-f002:**
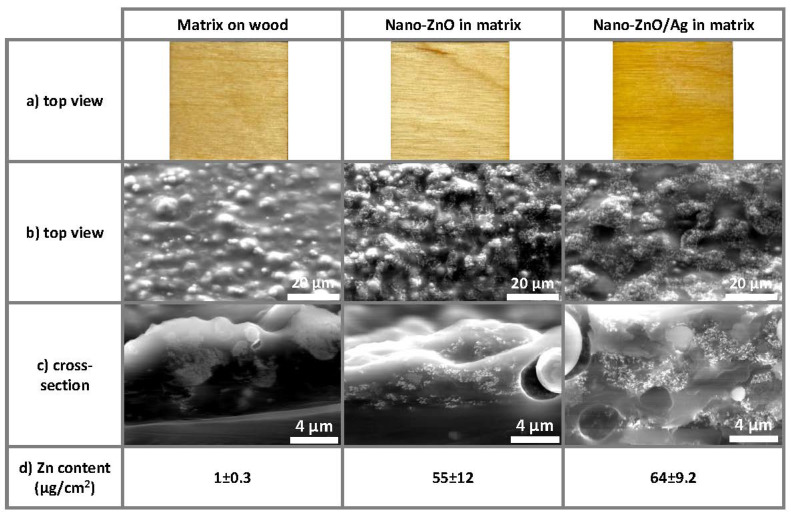
Photographic images, SEM images and XRF analysis data of coatings on wooden substrates: (**a**) photographic top view images of 25 × 25 mm samples reveal a more yellow coloring of nano-ZnO/Ag-based coatings compared to nano-ZnO-based and pure matrix coatings; (**b**) top view and (**c**) cross-section SEM images reveal a similar distribution for both nano-ZnO- and nano-ZnO/Ag-embedded nanoparticles; (**d**) Zn content (µg/cm^2^) was determined using X-ray fluorescence spectroscopy. Zn content in both nanoparticle-based coatings was similar and notably higher compared to the control (pure matrix-covered substrate).

**Figure 3 nanomaterials-11-03384-f003:**
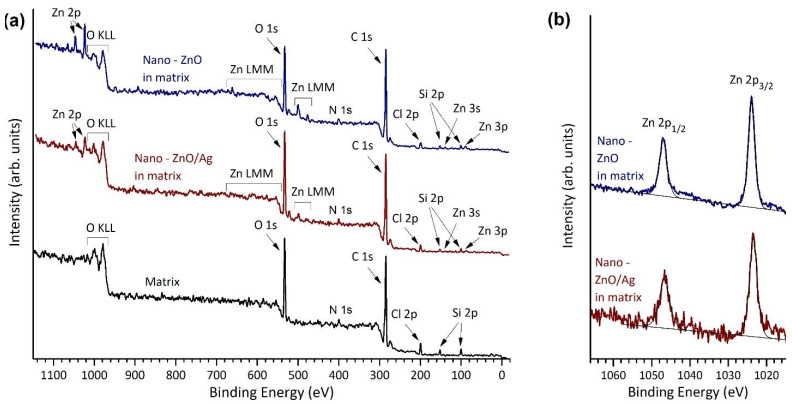
(**a**) XPS overview spectra (hv = 1486.6 eV, scan step 0.5 eV) and (**b**) Zn 2p XPS spectra (hv = 1486.6 eV, scan step 0.1 eV) of nano-ZnO- and nano-ZnO/Ag-containing coatings and pure matrix material on plywood substrates. According to the Zn 2p photoline positions and shape, Zn is in 2+-oxidation state.

**Figure 4 nanomaterials-11-03384-f004:**
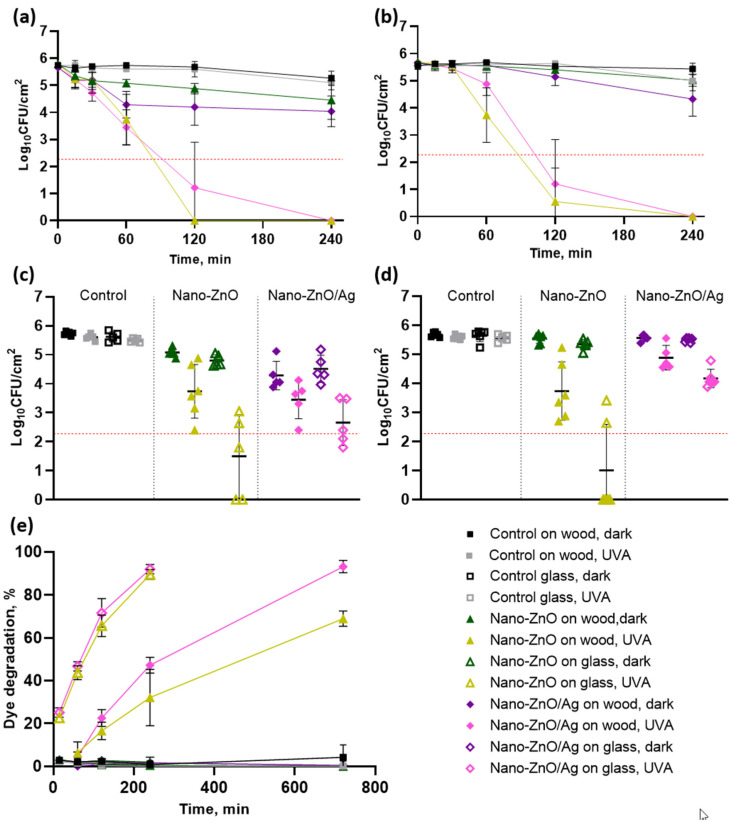
Antibacterial activity of matrix-embedded nano-coatings (on wood) (**a**,**b**) towards *E. coli* (**a**) and *S. aureus* (**b**); comparison of antibacterial effect of matrix-embedded nano-coatings (on wood) during 1 h and bare NPs on glass surfaces during 15 min (**c**,**d**) towards *E. coli* (**c**) and *S. aureus* (**d**). Photocatalytic dye degradation (**e**) is shown to illustrate differences in photocatalytic efficiency of the studied surfaces. Mean value from 3 experiments ±SD is presented. Red dotted line denotes the practical limit of quantification with 3 colonies counted in undiluted samples.

**Figure 5 nanomaterials-11-03384-f005:**
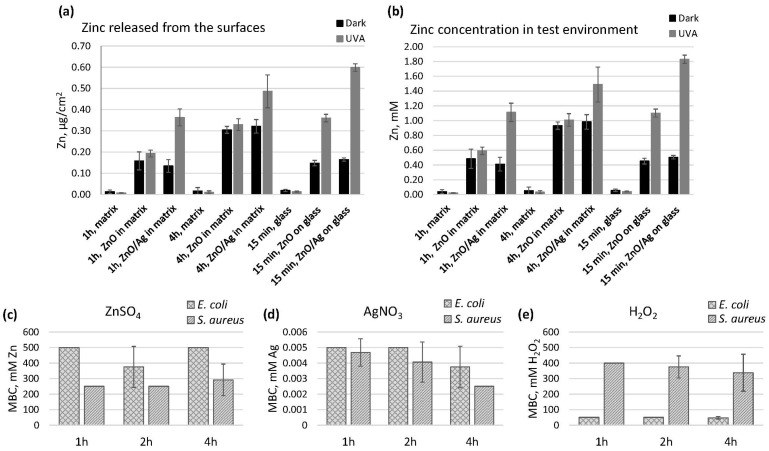
Zinc release from the nano-ZnO- and nano-Zno/Ag-based surfaces and minimal biocidal concentrations (MBC) of agents eliciting same or similar toxic effects to the mechanism of action of the nano-enabled surfaces. Release of Zn (**a**) and the corresponding Zn concentration in the test environment (**b**), as well as MBC values of Zn (**c**), Ag (**d**) and hydrogen peroxide (**e**) in the same test medium, are presented. UVA facilitated higher Zn release from surfaces containing nano-ZnO/Ag than from the ones without Ag. Release of Ag to the test environment was below the quantification limit of AAS and below MBC values (**d**). *S. aureus* was slightly more sensitive to the toxic effect of ionic Zn and less sensitive to ROS attack caused by peroxide than *E. coli*. Mean value of 3 experiments and at least 6 data points ±SD is presented.

**Figure 6 nanomaterials-11-03384-f006:**
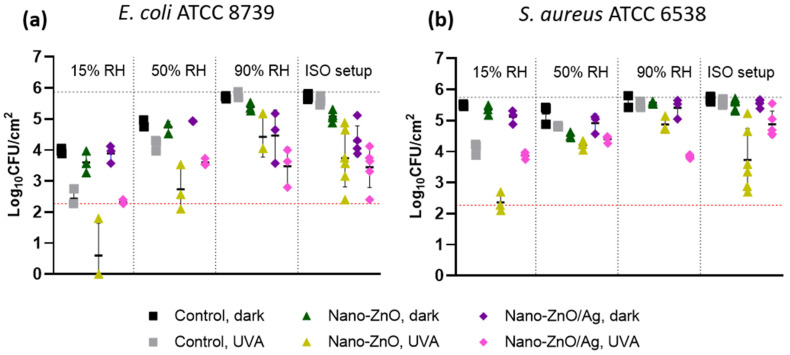
Antibacterial activity of surfaces based on matrix-embedded nano-ZnO or nano-ZnO/Ag towards *E. coli* (**a**) and *S. aureus* (**b**) in different relative air humidity (RH) conditions after 1 h. Nano-enabled surfaces generally demonstrate higher antibacterial ability at higher RH compared to the control surface in the same conditions; however, the effect of these surfaces is dependent on RH. Nano-Ag/ZnO surfaces lose their antibacterial activity at low RH in dark conditions (dark blue; **a**). UVA exposure is more toxic to both bacteria at lower RH and *E. coli* is more sensitive to drying at lower RH. Mean value of at least 3 experiments ±SD is presented. Red dotted line denotes the practical limit of quantification with 3 colonies counted in undiluted samples.

**Figure 7 nanomaterials-11-03384-f007:**
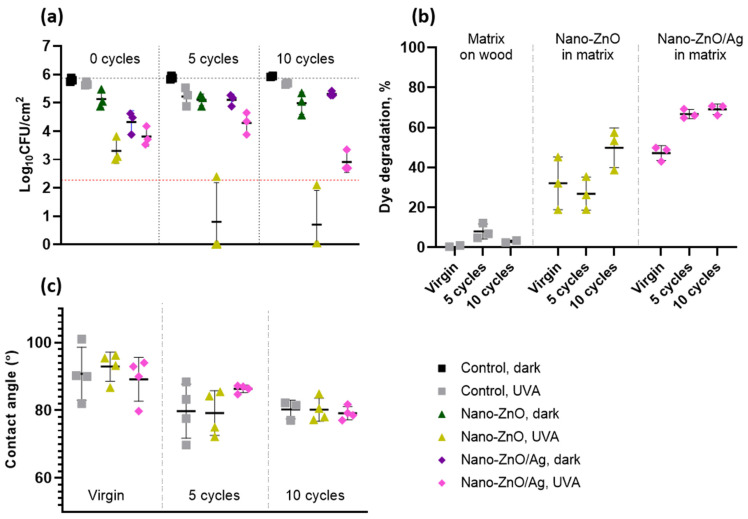
Effect of cyclic short-term wear on antibacterial activity towards *E. coli* (**a**) in the ISO test format, photocatalytic activity (**b**) and hydrophobicity (**c**) of the surfaces based on matrix-embedded nano-ZnO and nano-ZnO/Ag. Antibacterial activity of nano-ZnO/Ag surfaces in dark conditions decreased (+0.99 logs; *p* = 0.0056), while antibacterial activity of nano-ZnO (−1.0 logs; *p* = 0.0032) and nano-ZnO/Ag (−0.90 logs; *p* = 0.019) surfaces under UVA increased during 10 cycles of reuse compared to virgin surfaces (**a**). The latter can be explained by an increase in photocatalytic efficiency of nano-ZnO/Ag surfaces after 10 reuse cycles, with 18% (*p* = 0.0125) and 22% (*p* = 0.0025) increase in organic dye degradation (**b**). All the surfaces changed towards more hydrophilic surface properties, reflected in water contact angle measurements, which decreased ≥10% (*p* < 0.05) and stabilized after 10 cycles of reuse, while the number of reuse cycles explained 47% variation between samples (*p* < 0.0001) and surface type had no significant effect on hydrophilicity (**c**). Mean value of 3 experiments ±SD is presented. Red dotted line denotes the practical limit of quantification with 3 colonies counted in undiluted samples.

**Figure 8 nanomaterials-11-03384-f008:**
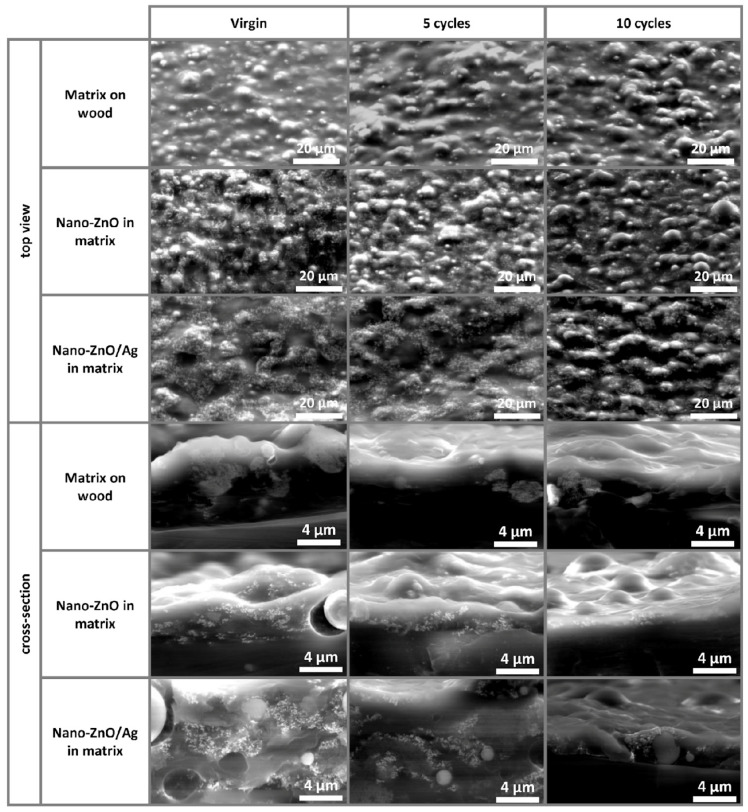
Scanning electron microscopy images of coatings on wooden substrates before (virgin) and after 5 and 10 reuse cycles. No apparent changes occurred to the coatings after reuse cycles and the embedded ZnO and ZnO/Ag nanoparticles were visible both in top view and cross-section images.

**Table 1 nanomaterials-11-03384-t001:** Atomic concentration of elements in nano-ZnO- and nano-ZnO/Ag-containing coatings and in pure matrix material on plywood substrates according to XPS analysis.

Element	C	O	Zn	Cl	Si	Ag	N
Nano-ZnO/Ag in matrix	70.6	21.3	0.5	1.6	3.5	not detected	2.5
Nano-ZnO in matrix	74.6	18.3	0.7	1.9	2.7	not measured	1.8
Matrix	69.6	23.4	not measured	2.4	3.5	not measured	1.1

## Data Availability

The data presented in this study are available on request from the corresponding author.

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
