# Peer review of "Preparation and Characterization of Photocatalytically Active Antibacterial Surfaces Covered with Acrylic Matrix Embedded Nano-ZnO and Nano-ZnO/Ag"

_nanomaterials, 2021, doi:10.3390/nano11123384_

Round 1

Reviewer 1 Report

The title does not reflect the content of the article. The photocatalytic properties of nano-ZnO and nano ZNO/Ag embedded in acrylic matrix
 is not clearly established. The manuscript entitled describes the fabrication and characterization of ZnO/Ag films following their antibacterial properties. Although the topic might be of interest for a broad range of readership, the manuscript fails to demonstrate the novelty of the experiment. The use of such films was previously demonstrated (authors admit this in introduction section). Additionally, it is well known that ZnO and Ag  have broad-spectrum activity against microorganisms. Silver inhibiting cellular respiration and disrupting metabolic pathways resulting in generation of reactive oxygen species disruption of DNA. Direct contact of ZnO-NPs with bacterial cell walls, resulting in destructing bacterial cell integrity and ROS formation (particulary under UV light). Moreover, the widespread use of silver-containing compounds has led to emergence of silver-resistant bacteria, so extending the applications to prevention of bacterial resistance is forced.

1. The same type of light activation was used  [ref. 17 entitled:  UVA-induced antimicrobial activity of ZnO/Ag nanocomposite covered surfaces]. The only novel element I can identify in the manuscript is a matrix. I find this aspect of interest, but unfortunately the manuscript is not following these aspects in depth.  How did the authors choose the thicknesses of coatings? It would be interesting if different layer thicknesses were tested. Moreover, the degradation of  methylene blue and contact angle mesurements were not described. 

2. I do not understand why authors used glass surface as nonomaterial control? 

Author Response

We thank the reviewer for his/her comments. These were very helpful in improving the manuscript. In the attached file, the comment or question of the reviewer is shown in blue and our response is written with black text.

Reviewer 2 Report

The manuscript ” Photocatalytic properties and antibacterial activity of surfaces covered with nano-ZnO and nano-ZnO/Ag embedded in acrylic matrix” in this paper, the authors prepared and analyzed the properties of antimicrobial coatings based on ZnO and ZnO / Ag nanoparticles embedded in the matte transparent acrylic coating layer, commercially available for wood surfaces.

The content of this study is comprehensive and needs major revision before it can be published in Nanomaterials.

Comments:

Photocatalytic activity depends on the crystal structure and intrinsic defects in materials. X-ray diffraction measurements should be carried out for all samples. Also, in order to highlighting potential intrinsic defects, electron paramagnetic resonance (EPR) spectroscopy should be carried out on the bare synthesized materials.

What was the pH value of the MB solution?

The degradation of the methylene blue occurs in the absence of the photocatalyst powders? The result must be introduced in the Fig 7b.

Author Response

(The authors gave the same response as above.)

Reviewer 3 Report

The manuscript describes the preparation of nano-ZnO and nano-ZnO/Ag materials embedded in an acrylic matrix and their use for the production of antibacterial surfaces. The antibacterial properties against E.Coli and S. Aureus have been investigated in dark conditions and under UVA light exposure, showing different effectiveness depending on the nanomaterials (ZnO or ZnO/Ag) and the environmental conditions (light, humidity, presence of matrix). The results are very dependent on the conditions for each material, but a general trend is not easily obtainable from the data shown here.

In my opinion the paper reports very interesting results, but some more information should be added in order to describe data more clearly. Minor revision is required, therefore, before publishing the paper.

Please, see comments below.

  • First of all, as a general comment, the colours chosen to display the data in figs.4(c,d,e), fig6 and fig.7 are too similar and it is difficult to distinguish the trends especially for dark and light green symbols. A different choise for the colours would facilitate readers to follow the discussion.

- Page 3 line 114: the reference samples on glasses have been annealed: which temperature? Does this modify the particles and, in particular, the interaction between ZnO and Ag in the nano-ZnO/Ag composite with respect to the unannealed sample? Indeed the annealed samples on glass show a faster photocatalytic dye degradation (fig.4e) and also a higher antibacterial activity (figs.4c and d) with respect to the same materials embedded in the matrix. Is this due only to a direct contact between nanomaterials and dye or bacteria? Do the author discard any effect of the annealing on the nanomaterials properties? Please comment about this in the text.

  • The improvement of the antibacterial properties is observed after 10 reuse cycles under UVA, but it does not seem to improve after 5 cycles with respect to the virgin material… why? A gradual improvement should be expected, shouldn’t it?
  • Furthermore, the improvement is probably due to a damaging of the matrix that leaves uncovered the active ZnO/Ag materials, even though the authors say that SEM does not confirm this. Do you have any other evidence of that? Apparently, a thinning of the coating is observed in fig.8 (cross sections) for nanoZnO and nanoZnO/Ag. Is it so? If the UVA exposure is carried on, will the matrix be completely destroyed by the photocatalytic process? This would not be good for applications.
  • As concerns the Zn release from the matrix, why is Ag found to enhance Zn release under UVA illumination?

Author Response

(The authors gave the same response as above.)

Round 2

Reviewer 1 Report

I have no additional comments.

Reviewer 2 Report

Based on the review provided by the authors, I would like to recommend the publication of the article, in its present form.

Reviewer 3 Report

The authors have suitably answered my questions and manuscript has been revised accordingly. In my opinion the paper can be published in the present form.